# The Effect of [(D-Ala^6^, Pro^9^NEt)mGnRH-a + Metoclopramide] (Ovopel) on Propagation Effectiveness of Two Breeding Lines of Common Carp (*Cyprinus carpio* L.) and on Luteinizing Hormone and 17α,20β-Dihydroxyprogesterone Levels in Females during Ovulation Induction

**DOI:** 10.3390/ani13081428

**Published:** 2023-04-21

**Authors:** Elżbieta Brzuska, Magdalena Socha, Jarosław Chyb, Mirosława Sokołowska-Mikołajczyk, Michał Inglot

**Affiliations:** 1Institute of Ichthyobiology and Aquaculture in Gołysz, Polish Academy of Sciences, Zaborze, Kalinowa 2, 43-520 Chybie, Poland; 2Department of Animal Physiology and Endocrinology, University of Agriculture in Krakow, Al. Mickiewicza 24/28, 30-059 Krakow, Poland; 3Department of Animal Nutrition and Biotechnology, and Fisheries, University of Agriculture in Krakow, Al. Mickiewicza 24/28, 30-059 Krakow, Poland

**Keywords:** common carp, Ovopel, luteinizing hormone, 17α,20β-dihydroxyprogesterone, reproduction

## Abstract

**Simple Summary:**

The aim of this study was to evaluate the effect of Ovopel on the reproductive effectiveness (measured as weight of eggs, egg quality, and ovulation rate) of females from two strains (Polish line 6 and Lithuanian line B) of common carp (*Cyprinus carpio* L.) and the release of luteinizing hormone (LH) and 17α,20β-dihydroxyprogesterone (17α,20β-DHP) during ovulation induction. After Ovopel treatment, the weight of eggs was higher in line 6, but the egg quality and ovulation rate were higher in line B. The observed differences in LH and 17α,20β-DHP levels between the lines were non-significant. LH levels between ovulated and non-ovulated fish did not differ within the lines. Steroid levels at 24 h after the priming dose were significantly higher in ovulated fish only in line 6. Summing up the results of comparing reproduction effects in two breeding lines of carp revealed higher reproduction effectiveness in B. Furthermore, the obtained results indicate that levels of tested hormones 12 h after the application of the resolving dose of Ovopel were higher in fish from that line which displayed higher reproduction effectiveness.

**Abstract:**

The study evaluates the impact of Ovopel on the reproductive effectiveness of carp from Polish line 6 and Lithuanian line B and the release of luteinizing hormone (LH) and 17α,20β-dihydroxyprogesterone (17α,20β-DHP) in females from these lines during ovulation induction. The levels of both hormones were determined in blood plasma samples taken just before the priming injection of Ovopel (0 h), at the time of administering the resolving dose of Ovopel (12 h), and after the next 12 h (24 h). Following Ovopel treatment, the mean egg weight obtained for line 6 was higher, but not statistically different, than that observed for line B. Egg quality, on the other hand, was significantly higher in line B. Female provenance did not significantly affect the number of eggs and living embryos after 70 h incubation. However, the total egg number for line 6 was higher. The mean number of living embryos (70 h) was similar for both lines. LH concentrations at 0, 12, and 24 h were not statistically different between the lines. A comparison of LH concentrations between ovulated and non-ovulated females at different sampling times revealed no significant differences either within or between the lines. Statistically significant differences in LH levels were found for both ovulated and non-ovulated females from a given line between the sampling times. The results for 17α,20β-DHP were similar, with only one difference: 24 h after the priming dose of Ovopel, 17α,20β-DHP levels in ovulated fish were significantly higher compared with non-ovulated females, but only in line 6.

## 1. Introduction

The common carp (*Cyprinus carpio* L.) is a fish species of significant importance for the aquaculture industries of many countries. In 2019, global carp production reached 3,821,611 tonnes [1]. In 2020, carp production in Poland was estimated at around 21,000 tonnes [2].

Even though carp can easily reproduce in captivity, spawning is not synchronized and usually occurs late in the growing season. This leaves only a short period for larval and fry training in their first year. In order to obtain as many fry as possible, carp are mainly bred under controlled conditions, not only in temperate countries. Such controlled reproduction involves the induction of ovulation using mainly carp pituitary homogenate/extract (CPH/CPE), e.g., [3,4,5,6,7,8] or mammalian/salmon gonadotropin-releasing hormone analog (GnRH-a) administered in combination with a dopamine antagonist, e.g., [9,10,11,12,13,14,15,16,17,18].

Mammalian gonadotropins such as human chorionic gonadotropin (hCG) and pregnant mare serum gonadotropin (PMSG), have not found wide application in the controlled breeding of carp, even though satisfactory propagation results can be achieved in this species with the administration of those heterologous gonadotropins to females (either alone or in combination with CPH) [19,20,21,22,23].

A series of studies on carp carried out at the Gołysz Institute (Poland) demonstrated that one cannot expect similar results of controlled reproduction in fish from different breeding lines after the induction of ovulation using preparations of natural origin (CPH, BPH, hCG, PMSG) and after the induction of ovulation with the use of synthetic preparations (GnRH-a), e.g., [6,13,14,15,21,24,25,26]. Differences in breeding results are observed between carp of different provenance even when the females used for breeding are of the same age and similar weight and are kept under the same conditions prior to the induction of ovulation, and when ovulation in females of various origins is induced using the same preparation and at the same time point [27]. Differences are usually observed between different lines in the ovulation ratio, weight, total number and quality of eggs, and, consequently, the number of living embryos and fry. The results of long-running studies on the stage of maturity of females from different breeding lines (as assessed by determining the degree of maturity of the oocytes of the oldest generation, sampled in vivo immediately prior to the administration of a spawning-inducing agent [28]) showed no significant difference in the stage of maturity between females of different provenance. 

In post-vitellogenic females, luteinizing hormone (LH—until 1999 called GtH) (whose synthesis and release from the pituitary gland is stimulated by the gonadotropin-releasing hormone—GnRH) stimulates the secretion from ovarian follicles of 17α,20β-dihydroxy4pregnen3 one (17α,20β-DHP), a maturation-inducing hormone—MIH [29,30,31,32]. 17α,20β-DPH induced the formation in oocytes of maturation-promoting factor (MPF), a complex of cdc kinase and cyclin B [33], which promotes the resumption of meiosis and subsequent ovulation. Thus, LH and 17α,20β-DPH are the main endocrine factors that regulate reproduction in post-vitellogenic females during the maturation of oocytes and ovulation [34,35].

The authors of the present study wondered whether the levels of those hormones present in the serum of female carp of different provenance at the time when ovulation induction begins and after the administration of a spawning-inducing agent may have a significant impact on the results of controlled reproduction. In order to answer the question, we conducted an experiment on carp from two genetically distant breeding lines (Polish line 6 and Lithuanian line B) forming part of the “live gene bank” of the Gołysz Institute of Ichthyobiology and Aquaculture, where we applied Ovopel as an ovulation stimulator, which is a complex preparation successfully used for the induction of ovulation and spermiation in carp and other important fish species, e.g., [36,37,38,39,40]. 

The aim of the present study was to characterize and compare the reproductive effectiveness of these two breeding lines, identify any significant differences in the LH and 17α,20β-DHP levels in the samples collected from females before and after Ovopel treatment between and within each of the lines studied, and determine whether differences in the levels of LH and 17α,20β-DHP between ovulated and non-ovulated females from the same line at a particular sampling time were significant.

## 2. Materials and Methods

### 2.1. Ethics

The experiment was conducted in accordance with the principles of the Ethical Committee for the Protection of Research Animals at the Polish Academy of Sciences. The study was approved by the II Local Institutional Animal Care and Use Committee (IACUC) in Kraków, Poland (resolution No. 83/2019). 

### 2.2. Handling of Spawners before the Induction of Ovulation and Treatments 

The experiment was conducted at the Institute of Ichthyobiology and Aquaculture in Gołysz (Polish Academy of Sciences) during the natural spawning period of carp (in May). The study included 28 females aged 10–11 years, weighing 7.1–11.1 kg. These included 13 females from Lithuanian line B and 15 females from Polish line 6 [41]. The fish were sampled based on external morphological signs of maturity (well-rounded soft abdomen and swollen genital opening [7,42]) from a large population harvested from the pond (in spring, when water temperature reached 18 °C). The females were transported to the hatchery and placed in seven 3 m^3^ tanks (4 females per tank). After a 2-day period of acclimatization, during which the water temperature was gradually increased from 18–19 °C to 20–21 °C, the females received a priming dose (1/5 pellet kg^−1^ BW) of Ovopel [D-Ala^6^, Pro^9^NEt-mGnRH-a) + metoclopramide] (Interfish, Budapest, Hungary). After 12 h, a resolving dose of Ovopel of 1 pellet kg^−1^ of the female’s BW was administered to the females. One pellet of Ovopel contains 18–20 µg of D-Ala^6^, Pro^9^NEt-mGnRH-a, and 8–10 mg of metoclopramide. Ovopel pellets, just like dried pituitary glands, were ground in a mortar and then used to prepare a suspension in saline [43]. Both doses were administered by intraperitoneal injection. In order to increase the yield and quality of the sperm used for fertilization, males from line B and line 6 received an intraperitoneal dose of Ovopel of 1 pellet kg^−1^ BW.

A heparinized 21-gauge needle with a 1 mL syringe was used to collect serial blood samples (1000 μL) from the females by caudal venipuncture. The first sample was collected prior to the first injection of Ovopel (time 0 h), the second at the time of the second injection of Ovopel (time 12 h), and the third sample was collected at the time of fish ovulation checked at approximately 12 h after the second injection of Ovopel (time 24 h). The blood samples were centrifuged at 4000× *g* for 5 min at 8 °C, and plasma was stored at −80 °C until analysis.

### 2.3. Handling of Eggs Obtained

Females were checked for ovulation every 1 h as from 12 h post the resolving dose of Ovopel until 20 h after this dose. Eggs released in response to gentle abdominal pressure were stripped into a dry plastic container. The eggs obtained from each female were weighed and fertilized with pooled sperm collected from 3–4 males from the same breeding line as the females that had released the eggs. After the elimination of stickiness using the method described by Woynarovich and Woynarovich [44], 300 g of eggs from each female were incubated in separate Weiss glasses with a capacity of 7 L in water at a temperature of 21 ± 1 °C. After 24, 48, and 70 h of incubation, the percentage of living embryos was calculated for each female. In addition, the total number of eggs, taking into consideration the weight of 1 carp egg depending on the female BW class (as described by Cejko and Brzuska [45]) and the number of living embryos after 70 h of incubation were calculated for each fish. The percentage of ovulating females was also calculated for the lines studied and the latent period.

### 2.4. Hormone Analysis

LH and 17α,20β-DHP concentrations in plasma samples were assayed using the ELISA method: LH as described by Kah et al. [46] and 17α,20β-DHP as described by Szczerbik et al. [47]. LH and 17α,20β-DHP concentrations were determined by measuring the absorbance using a 96-well plate reader (BIO-TECH INSTRUMENTS, EL 311) at 490 and 450 nm, respectively.

### 2.5. Statistical Analysis

#### 2.5.1. Statistical Analysis Relating to the Reproduction Effects

Basic statistical tests of the data were carried out using standard procedures included in the Statistica 10 software (StatSoft Polska Sp. Z o.o.). For data expressed as percentages, transformations were made using the arcsine function before the analysis. Data on the breeding were analyzed using the least-squares analysis of variance [48] in order to determine the impact of the females’ provenance on the variables analyzed. The variables included the weight of eggs in grams, the weight of eggs expressed as a percentage of the female’s body weight, the percentage of living embryos after 24, 48, and 70 h of incubation, the total number of eggs, and the number of living embryos after 70 h of incubation. 

The analysis of variance was performed according to the following linear model:*Y_ij_* = *α* + *t_i_* + *bW_ij_* + *e_ij_*
where *Y_ij_* is observation… *j*, *α* is the theoretical overall mean with the assumption that *W_ij_* = 0, *t_i_* is the effect of the female’s provenance, *i* = 1, 2…, *b* is partial regression in the female’s body weight, *W_ij_* is the female’s body weight, and *e_ij_* is the random error associated with observation *j*.

The analysis allowed for estimating least-squares means for the variables within each breeding line. The significance of the impact of females’ provenance on the variables determining reproductive effectiveness was tested using the F-test.

As not all females from lines B and 6 released eggs at the same time following the administration of the resolving dose of Ovopel, the question arose whether the reproduction performance of fish that had released eggs earlier differed from the reproduction performance of those fish that had released eggs later. In order to answer this question, a least-squares analysis of variance was performed for each line, in which the classification factor was the latent period. The analyses were performed according to the following linear model:*Y_ij_* = *α* + *l_i_* + *bW_ij_* + *e_ij_*
where *Y_ij_* is observation… *j*, *α* is the theoretical overall mean with the assumption that *W_ij_* = 0, *l_i_* is the effect of the latent period, *I* = 1, 2…, *b* is partial regression on the female’s body weight, *W_ij_* is the female’s body weight, and *e_ij_* is the random error associated with observation *j*.

The analysis allowed for estimating least-squares means for the variables determining reproduction effectiveness when the latent period differed both within line B and within line 6.

The authors of the present study also examined whether the difference in reproduction effects between line B and line 6 was significant when the eggs in both lines were obtained at the same latent period. In order to answer this question, the least squares analysis was performed according to the linear model in which the main classification factor was the line. The significance of the impact of the latent period within line B and line 6 and of the line at the same latent period on the variables determining propagation effects was tested using the F-test. An X^2^ (chi-square) test was used to verify the dependence of distribution on classification (lines B and 6; ovulated and non-ovulated females) [49].

A multiple regression equation was formulated for each line to predict the number of living embryos after 70 h of incubation. In this equation, the weight of females, the weight of eggs in grams, and the weight of eggs expressed as a percentage of the female’s BW were used as independent variables.

#### 2.5.2. Statistical Analysis Relating to Hormone Analysis

The resulting LH and 17α,20β-DHP concentrations were analyzed using GraphPad Prism statistical software (version 5, 2007, GraphPad Software: San Diego, CA, USA). All data are presented as arithmetical means ± standard error of means (SEM). A non-parametric two-tailed Mann–Whitney U-test was performed for the comparison of hormone concentrations in fish from different lines. In the case of hormone levels measured over time within a given line of fish, the Friedman test followed by Dunn’s test was performed. The differences between the means were considered significant for *p* ≤ 0.05.

## 3. Results

### 3.1. Reproductive Performance 

#### 3.1.1. Female Ovulation after Ovopel Treatment and Latent Period

In females from Lithuanian line B and those from Polish line 6, ovulation occurred at two time points. In 30.7% of females from line B, ovulation occurred 13 h after administration of the resolving dose of Ovopel, whereas in 46.2% of females from this line, ovulation occurred after a further 2 h. In turn, 33.3% and 20% of females from line 6 released eggs 15 h and 18 h after the second injection of Ovopel, respectively.

The ovulation ratio for line B was 10/13 and for line 6 8/15. The X^2^ test performed for the classification adopted and the distribution obtained did not reveal any statistically significant differences.

#### 3.1.2. Effect of the Provenance of Females on the Weight and Quality of Eggs Obtained

The provenance of females did not significantly affect the weight of eggs obtained, either when expressed in grams or as a percentage of the female’s body weight. However, the least-squares means estimated for those variables clearly show that the weight of eggs obtained from females from Polish line 6 was higher by 303 g and 2.95%, respectively, compared with the weight of eggs released by females from line B (Table 1). The provenance of females significantly determined (*p* ≤ 0.05) the percentage of living embryos after 24, 48, and 70 h of incubation. The means estimated for those variables were higher for line B in the case of all three periods analyzed (Table 1). The percentage of living embryos after 70 h of incubation for females from line B was higher by as much as 14.5% than that calculated for line 6 (Table 1).

#### 3.1.3. Effect of the Provenance of Females on the Total Number of Eggs and the Number of Living Embryos after 70 h of Incubation

The effect of the line of the females was statistically insignificant in respect of the total number of eggs and the number of living embryos after 70 h of incubation. However, the least-squares means estimated for the variable clearly show that the number of eggs obtained from females from line 6 was higher by 209,000. The number of eggs for this line was higher by 104,000 than the mean value for the entire set (Table 1). The mean number of living embryos (70 h) was similar for both the lines studied (Table 1).

#### 3.1.4. Latent Period and the Weight and Quality of Eggs Obtained

Ovulation time did not significantly determine the weight of eggs obtained from females from line B. However, it should be stressed that the weight of eggs released by females 13 h after administration of the resolving dose of Ovopel was higher than the weight of eggs obtained from those females from this line that ovulated 2 h later (Table 2). The percentage of living embryos after 24, 48, and 70 h of incubation was similar for both latent periods in females from line B (Table 2).

Similarly, no statistically significant association was found between the latent period and the weight of eggs obtained from females from line 6. However, it was observed that the mean weight of eggs obtained from these females 15 h after administration of the second dose of Ovopel was higher by 351 g compared with the mean estimate for the weight of eggs released by the fish which ovulated 3 h later. The latent period was a significant determinant of the quality of eggs produced by females from line 6. The percentage of living embryos after 24, 48, and 70 h of incubation was higher for the latent period of 15 h (Table 2). It should be stressed that both after 48 h and after 70 h of incubation, the percentage of living embryos in eggs obtained from females from this line after longer latency was lower by as much as 34% (Table 2).

A comparison of the mean weight of eggs (g) released by females from line B and females from line 6 15 h after the second injection of Ovopel showed that the mean calculated for this variable was significantly higher for line 6. The mean weight of eggs expressed as a percentage of the female’s body weight was also significantly higher for line 6 (Table 2). However, the quality of eggs from females from this line after 48 h and after 70 h of incubation was substantially poorer (Table 2).

#### 3.1.5. Latent Period and the Total Number of Eggs and the Number of Living Embryos after 70 h of Incubation

A significant effect of ovulation time on the total number of eggs and the number of living embryos (70 h) was noted only for line B (Table 2). The least-squares mean estimated for the total number of eggs was higher for the latent period of 13 h. The number of eggs obtained from females from line 6 15 h after administration of the resolving dose of Ovopel was lower compared with the number of eggs obtained from those females that ovulated 3 h later. However, the number of living embryos (70 h) was higher by 90,000 in the case of the latent period of 15 h (Table 2).

Analyzing the least-squares mean values for the number of eggs and the number of living embryos (70 h) in the case of the latent period of 15 h revealed that the mean for those variables was significantly higher for line 6 (Table 2).

### 3.2. Regression Predictions

Table 3 shows multiple regression equations for the number of living embryos after 70 h of incubation of eggs obtained from females from line B and females from line 6, as well as the values of the coefficient of determination (R^2^). The data presented show that the predictability of this trait was more satisfactory for line B (R^2^ = 0.98).

### 3.3. Results of Hormone Level Analysis

#### 3.3.1. LH Levels in Fish from Both Lines Sampled at 0, 12, and 24 h 

In each line of fish, the levels of LH increased significantly over time (Figure 1): in line 6 from 2.798 (0 h) to 200.3 ng mL^−1^ (24 h) (*p* ≤ 0.05) and in line B from 2.669 (0 h) to 242.7 ng mL^−1^ (24 h) (*p* ≤ 0.05). The comparison of LH concentrations between the lines within each sampling time did not reveal any significant differences. However, the mean LH level at 24 h sampling time in females from line B was 42.4 ng mL^−1^ higher than in fish from line 6.

#### 3.3.2. LH Levels in Ovulated or Non-Ovulated Fish from Both Lines Sampled at 0, 12, and 24 h

There were no statistically significant differences in LH concentrations between ovulated and non-ovulated fish within the lines and sampling times (Figure 2). However, at 12 and 24 h after the priming dose of Ovopel LH concentrations in ovulated fish from line 6 were lower than in non-ovulated ones, which stands in contrast to line B, where LH levels in ovulated females were higher than in non-ovulated fish. 

The same data presented over time (Figure 3A,B) showed that within each line the levels of LH increased significantly for both ovulated and non-ovulated fish: in ovulated females from line 6 from an average of 2.65 ng mL^−1^ before the first injection of Ovopel to 191.94 ng mL^−1^ at 24 h post-injection (*p* ≤ 0.05); in line B from 3.045 ng mL^−1^ before injection to 245.46 ng mL^−1^ post-injection (*p* ≤ 0.05) (Figure 3A). 

In females which did not ovulate (Figure 3B), LH levels also increased significantly over time: in fish from line 6 from 2.95 ng mL^−1^ (before treatment) to 209.78 ng mL^−1^ at 24 h post the first Ovopel injection (*p* ≤ 0.05). In non-ovulated females from line B, LH concentrations were significantly higher (*p* ≤ 0.05) at 24 h after the priming dose of Ovopel (an increase from 1.41 ng mL^−1^ at 0 h to 233.29 ng mL^−1^).

#### 3.3.3. 17α,20β-DHP Levels in Fish from Both Lines Sampled at 0, 12, and 24 h

In each line of fish, the levels of 17α,20β-DHP increased significantly over time (Figure 4): in fish from line 6 from 0.1918 ng mL^−1^ before the first injection of Ovopel (0 h) to 0.5481 ng mL^−1^ 24 h later and in line B from 0.189 ng mL^−1^ to 1.359 ng mL^−1^ (*p* ≤ 0.05). 

The comparison of 17α,20β-DHP concentrations between the lines within each sampling time did not show statistically significant differences. However, the mean 17α,20β-DHP level at 24 h sampling time in females from line B was 0.811 ng mL^−1^ higher than in females from line 6 (Figure 4).

#### 3.3.4. 17α,20β-DHP Levels in Ovulated or Non-Ovulated Fish from Both Lines Sampled at 0, 12, and 24 h

There were no statistically significant differences in 17α,20β-DHP concentrations between ovulated and non-ovulated fish within the lines before the first injection of Ovopel and 12 h post-injection (Figure 5). At 24 h sampling time, the level of steroid concentrations in ovulated fish from line 6 (0.664 ng mL^−1^) was significantly higher than in non-ovulated ones (0.416 ng mL^−1^) (*p* ≤ 0.05). At the same sampling time in fish from line B, steroid levels were 1.639 ng mL^−1^ and 0.423 ng mL^−1^ in ovulated and non-ovulated females, respectively, but this difference was not statistically significant.

The same data presented over time (Figure 6A,B) showed that within both lines of ovulated females, the levels of 17α,20β-DHP increased significantly: from 0.1958 ng mL^−1^ at 0 h to 0.6641 ng mL^−1^ at 24 h in fish from line 6 (*p* ≤ 0.05) and from 0.199 ng mL^−1^ to 1.639 ng mL^−1^ in fish from line B (*p* ≤ 0.05) (Figure 6A). In ovulated fish from line B, a statistically significant (*p* ≤ 0.05) difference between the levels of 17α,20β-DHP at sampling times 0 h and 12 h was noted. In non-ovulated fish (Figure 6B) from line 6, a significant (*p* ≤ 0.05) increase in the steroid level was found at 24 h after the first injection of Ovopel as compared with the initial level (0 h). In fish from line B, there were no differences in steroid levels between sampling times.

## 4. Discussion

Based on the results obtained in the present study, it can be noted that the reproduction effectiveness differed between the carp lines investigated. Better reproduction effects were demonstrated for line B, even though the weight and the total number of eggs released by females from line 6 were higher.

The percentage of ovulated females from line B was higher as compared to all the fish of this origin undergoing hormonal ovulation induction. Furthermore, within this line, there were no such substantial losses during the egg incubation time as were observed in line 6. The number of living embryos (70 h) was similar for both lines (slightly above 500,000) but the predictability of this variable was more precise for line B. From the breeding practice perspective, it is important that the costs incurred for inducing ovulation and obtaining a specific number of living embryos should be significantly lower in the case of line B. The latent period observed for the two lines also differed. Females from both line B and line 6 released eggs at two time points, but both these latent periods (earlier and later) were shorter for line B than those for line 6. This indicated that females from the Lithuanian line matured faster after the administration of the resolving dose of Ovopel. Within line B, eggs were stripped from a larger percentage of ovulated females and the number of living embryos (70 h) was higher after the longer latent period (15 h) as compared to the corresponding values after the latent period shorter by 2 h. On the other hand, within line 6 the percentage of ovulated females and the number of living embryos (70 h) were higher in females which released eggs after the shorter latent period. Higher reproduction effectiveness in line B compared to line 6 was also noted in studies reported by Brzuska [24] and Cejko & Brzuska [26], not only after Ovopel application, but also after twofold hypophysation. 

To date, research conducted at the Gołysz Institute on the effectiveness of reproduction in carp from different breeding lines has not taken into account the profile of hormonal changes in spawners during the ovulation induction period. Data about changes in the levels of such hormones as gonadotropins or steroids in response to the use of ovulation-stimulating agents are significant for knowledge-building reasons, but they can also have some importance in the context of reproduction outcomes in this important fish species.

The subject literature includes numerous works demonstrating the effects of carp pituitary extract or GnRH-a on plasma gonadotropin release and/or on concentrations of gonadal steroids associated with oocyte maturation during spawning induction in the carp, e.g., [29,50,51,52,53,54,55]. However, the authors of the present study are not aware of any publications addressing the problem of the association between the levels of these hormones measured during ovulation induction in common carp and the effects of controlled reproduction characterized by numerous traits. It is worth noting that changes in GtH levels in parallel with the progress of oocyte maturation were reported in common carp treated with LHRH by Sokołowska [56], with LHRH or LHRH-a by Billard [51], with CCPE by Levavi–Zermonsky and Yaron [57], and in fish from this species treated with an sGnRH superactive analog combined with/without dopamine receptor antagonist metoclopramide or with CCPE by Drori et al. [58].

The study described in the present paper demonstrated that the average LH concentrations in groups that included all females within the line and those observed after dividing the fish into ovulated and non-ovulated did not differ statistically significantly between the females from the lines investigated. The only difference was observed at 24 h sampling time (i.e., 12 h after the application of the resolving dose of Ovopel, which contained 18–20 µg of D-Ala^6^,Pro^9^NEt-mGnRH-a and 8–10 mg of metoclopramide), with LH levels in females from line B clearly, however, not statistically, higher than in fish from line 6. It is interesting to note that within line 6, the mean concentrations of LH in samples collected both 12 h and 24 h after the application of the priming dose of Ovopel were higher in non-ovulated fish, whereas within line B higher LH concentrations at both these time points were observed in ovulated fish.

Billard et al. [51] reported the plasma GtH profile in ovulated and non-ovulated carp females receiving two injections of LHRH-A (5 µg and 50 µg of female’s BW). These authors note that the plasma GtH levels in ovulated fish ranged from 20 to 58 ng mL^−1^ between 6 h and 18 h after the first injection of LHRH-A, but the mean levels in the ovulated females were not significantly different from the levels in non-ovulated fish treated with LHRH-A.

Our study did not demonstrate a significant difference in the basal levels of LH between the lines. However, these levels were low for both lines (exceeding 2.5 ng mL^−1^, but not reaching 3 ng mL^−1^) when compared with corresponding data from experiments on carp conducted by other authors, e.g., [54,55,58,59].

Comparing classical hypophysation with LHRH application in females of two strains of carp (strain Z and strain S), Weil et al. [60] found that even though the GtH level prior to the first injection of CPE was significantly different for these fish strains (Z—6.36 and S—3.24 ng mL^−1^), the sensitivity of fish of different provenance to hypophysation was similar—the same pattern of changes in the GtH level was observed for each of the lines.

The analysis of the LH levels during our experiment showed a similar hormonal pattern of LH secretion for both lines and for ovulated and non-ovulated females. At 12 h after the first injection, the rise in LH concentrations averaged 40 ng mL^−1^ and was not significantly different in relation to the pre-injection values. During the next 12 h after the first injection (24 h of the experiment), the average LH level was about 220 ng mL^−1^, which means an increase by an average of 180 ng mL^−1^. This increase was statistically significant in relation to the initial concentrations (levels measured before the first Ovopel administration) but not significant in comparison to the levels found 12 h later.

Similar to the LH levels, the concentrations of 17α,20β-DHP did not show significant differences between the lines of fish when a comparison was made between ovulated and non-ovulated females. When comparing the levels of this steroid between ovulated and non-ovulated groups of both lines, a statistically significant difference was observed between ovulated and non-ovulated females but only within 24 h of the first Ovopel injection and only in line 6. The level of this steroid was significantly higher in the group of ovulated females.

The analysis of 17α,20β-DHP levels during our experiment showed that 12 h after the first injection of Ovopel, the steroid level increased by an average of 0.33 ng mL^−1^ (a similar increase in both lines and similar for ovulated and non-ovulated fish). This increase was statistically significant compared to the baseline value only for line B in ovulated fish. At 24 h after the first Ovopel injection in ovulated females of both lines and non-ovulated ones from line 6, the steroid levels were significantly higher in comparison with its initial concentrations.

Weil et al. [61] determined the level of 17α,20β-DHP depending on the occurrence or lack of ovulation in carp submitted to hypophysation and found that in non-ovulated fish the level of this steroid was in most cases undetectable (<0.5 ng mL^−1^). In partially ovulated females, there was a slow increase in the level of this hormone between 18 h and 21 h after the CPE injection, which was followed by a decrease in the level of this steroid after 24 h. In ovulated fish, the level of 17α,20β-DHP was high (7 ng mL^−1^), though not as high as that (111 ng mL^−1^) reported by Levavi-Zermonsky and Yaron [57]. In our study, 24 h after administering the priming dose of Ovopel, the level of 17α,20β-DHP remained significantly higher than the concentration of this hormone prior to the injection in ovulated fish from both lines and in non-ovulated fish from line 6. A somewhat different profile of this steroid was observed by Peter et al. [62] in a study on goldfish. These authors demonstrated a significant increase in the levels of both gonadotropin and 17α,20β-DHP as soon as 6 h after the application of a GnRH-a and pimozide. After a further 14 h, the level of gonadotropin remained significantly higher as compared to the control, whereas the level of 17α,20β-DHP decreased dramatically, down to the concentrations observed in the control group. In their experiment involving the application of sGnRH-a with metoclopramide to female carp, Drori et al. [58] reported that the level of 17α,20β-DHP started increasing 7 h after the injection of this analog, reaching 23.9 ng mL^−1^ after another 4 h. Twenty-six hours after administering sGnRH-a + metoclopramide, the level of this steroid decreased down to the baseline value.

Based on the results of our research, it may be concluded that the profiles of the two hormones investigated in the blood serum of females from line 6 and line B sampled before and after Ovopel administration generally do not deviate from the results obtained by other authors in similar experiments on carp [34,60]. Nevertheless, there are some differences: in our experiment, 24 h after the application of the priming dose of Ovopel, the levels of both LH and 17α,20β-DHP did not show a downward trend, as was the case in the above-cited studies. Furthermore, the levels of 17α,20β-DHP determined in our study are relatively low, with the highest values not exceeding 2 ng mL^−1^. 

It seems justified to mention that when analyzing the data obtained in the course of the study, an association emerged between the levels of LH and 17α,20β-DHP during ovulation induction and selected features describing the effectiveness of reproduction in the breeding lines studied. Our data on the concentration of these hormones 12 h after the application of the resolving dose of Ovopel show that the levels of both hormones were higher for line B. Eggs were obtained from a higher percentage of females from this line and their quality was considerably higher. The quality of eggs, measure as the ability to be fertilized and subsequently develop into a normal embryo is the most important issue in aquaculture [63,64]. Further research, also at the molecular level (the eventual alterations in the transcript of reproductive-related genes), is needed to explain the observed differences in reproductive effectiveness between studied strains of common carp.

## 5. Conclusions

The results of comparing reproduction effects in two breeding lines of carp, i.e., line 6 and line B, revealed higher reproduction effectiveness in B. The differences in LH and 17α,20β-DHP serum levels in these lines were not statistically significant, either in samples collected just before administering the priming dose of Ovopel, 12 h after administering the priming dose (i.e., at resolving dose application) or 12 h after administering the resolving dose. However, the results obtained clearly indicate that concentrations of each of these hormones 12 h after the application of the resolving dose of Ovopel were higher in fish from that line which displayed higher reproduction effectiveness. Further studies are needed, e.g., measured by changes in mRNA transcript abundance of genes essential for reproduction, to explain these differences in reproductive effectiveness between female carp of different provenance. 

## Figures and Tables

**Figure 1 animals-13-01428-f001:**
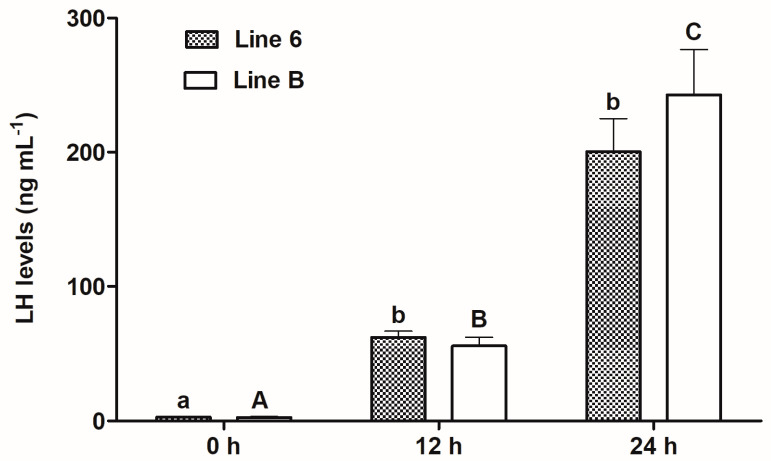
Plasma LH levels in blood samples taken from females from two breeding lines of common carp, i.e., Polish line 6 (*n* = 15) and Lithuanian line B (*n* = 13) before the first injection of Ovopel (0 h), 12 h after this injection, i.e., at the time of the second injection of Ovopel (12 h) and 24 h after the first injection of Ovopel (i.e., 12 h after the second injection) (24 h). Different lowercase letters indicate significant differences between sampling times within line 6. Different uppercase letters indicate significant differences between sampling times within line B (*p* ≤ 0.05). Differences between lines within sampling times were non-significant.

**Figure 2 animals-13-01428-f002:**
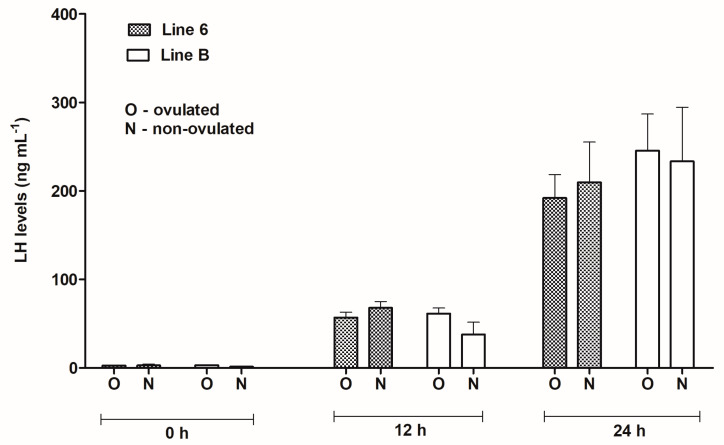
Plasma LH levels in blood samples taken from ovulated and non-ovulated females from Polish line 6 and Lithuanian line B before the first injection of Ovopel (0 h) and 12 and 24 h post-injection (12 and 24 h).

**Figure 3 animals-13-01428-f003:**
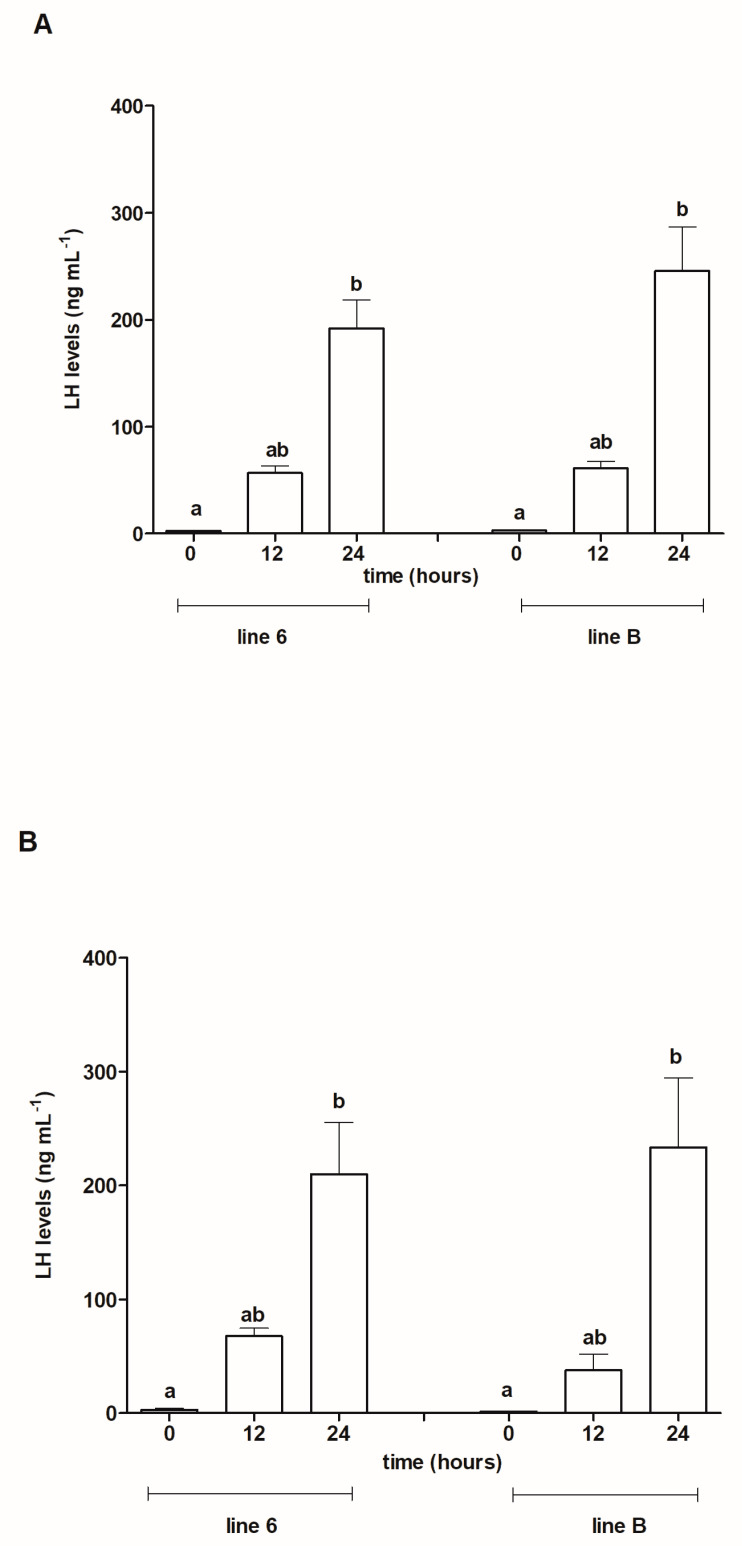
Plasma LH levels in blood samples taken from ovulated females from Polish line 6 and Lithuanian line B (**A**) and non-ovulated females from these lines (**B**) before the first injection of Ovopel (0 h) and 12 and 24 h post-injection (12 and 24 h). Different letters with ovulated and non-ovulated females of each line denote statistically significant differences between sampling times (*p* ≤ 0.05).

**Figure 4 animals-13-01428-f004:**
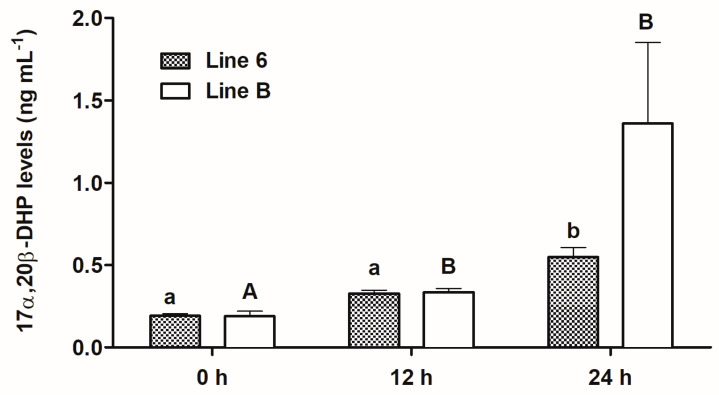
Plasma 17α,20β-DHP levels in blood samples taken from females from two breeding lines of common carp, i.e., Polish line 6 (*n* = 15) and Lithuanian line B (*n* = 13) before the first injection of Ovopel (0 h) and 12 and 24 h post-injection (12 and 24 h). Different lowercase letters indicate significant differences between sampling times within line 6. Different uppercase letters indicate significant differences between sampling times within line B (*p* ≤ 0.05). Differences between lines within sampling times were non-significant.

**Figure 5 animals-13-01428-f005:**
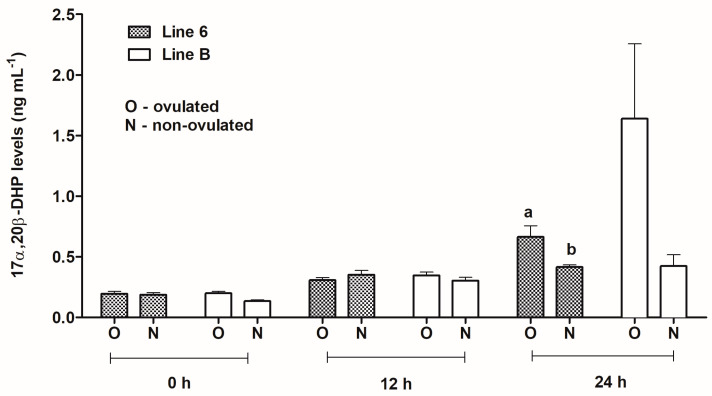
Plasma 17α,20β-DHP levels in blood samples taken from ovulated and non-ovulated females from Polish line 6 and Lithuanian line B before the first injection of Ovopel (0 h) and 12 and 24 h post-injection (12 and 24 h). Different letters indicate significant differences between the levels of steroid concentrations in ovulated and non-ovulated fish within line 6 at 24 h sampling time (*p* ≤ 0.05).

**Figure 6 animals-13-01428-f006:**
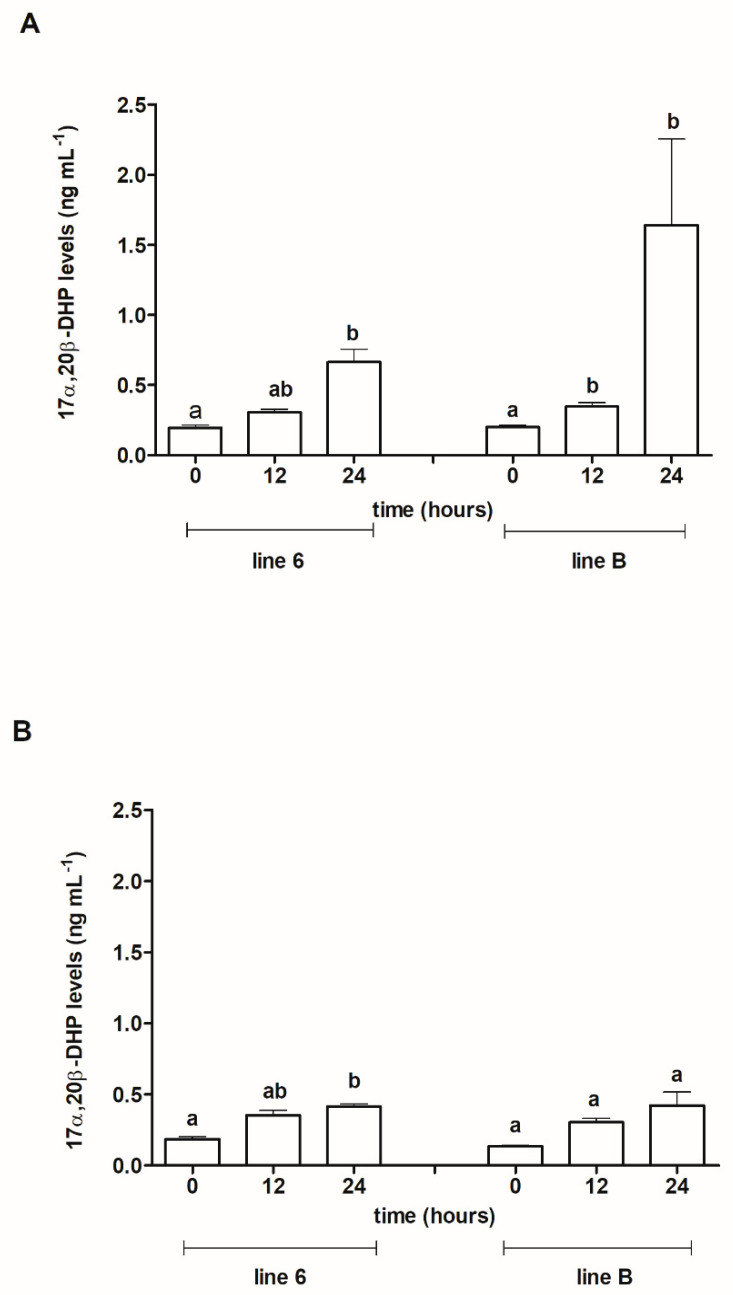
Plasma 17α,20β-DHP levels in blood samples taken from ovulated females from Polish line 6 and Lithuanian line B (**A**) and non-ovulated females from these lines (**B**) before the first injection of Ovopel (0 h) and 12 and 24 h post-injection (12 and 24 h). Different letters used with ovulated females from both lines and non-ovulated females from line 6 denote statistically significant differences between sampling times (*p* ≤ 0.05). Levels in non-ovulated fish from line B designated by the same letter do not differ significantly from each other (*p* ≤ 0.05).

**Table 1 animals-13-01428-t001:** Constants (LSC) and least-squares means (LSM) estimated for the traits determined propagation results of breeding lines B and 6 and results of F-test.

	Investigated Traits															
Classification factor	Weight of eggs (g)				Weight of eggs (% of female’s body weight)				Percentage of living embryos							
									after 24 h incubation				after 48 h incubation			
	α = 1095.10				α = 10.75				α = 85.66				α = 79.99			
	LSC	LSM	Se	F	LSC	LSM	Se	F	LSC	LSM	Se	F	LSC	LSM	Se	F
Breeding line				-				-				*				*
Lithuanian B	−151.35	943.35	194.16		−1.48	9.27	1.77		4.27	89.93	2.89		7.60	87.59	4.02	
Polish 6	151.35	1246.44	189.19		1.48	12.22	1.72		−4.27	81.38	2.91		−7.60	72.38	3.92	
	Percentage of living embryos after 70 h incubation				Number of eggs (in thousands)				Number of living embryos after 70 h incubation (in thousands)							
	α = 73.22				α = 740.8				α = 529.1							
	LSC	LSM	Se	F	LSC	LSM	Se	F	LSC	LSM	Se	F				
Breeding line				*				-				-				
Lithuanian B	7.25	80.47	4.36		−104.60	636.2	129.8		−6.07	523.2	101.4					
Polish 6	−7.25	65.97	4.25		104.60	845.4	126.4		6.07	535.2	98.8					

Se—standard error of the LSM; * *p* ˂ 0.05.

**Table 2 animals-13-01428-t002:** Constants (LSC) and least-squares means (LSM) characterizing the reproductive effectiveness of two breeding lines of common carp (*Cyprinus carpio* L.) associated with the latent period and results of the F-test.

	Investigated Traits															
Classification factor	Weight of eggs (g)				Weight of eggs (% of female’s body weight)				Percentage of living embryos after 24 h incubation							
	LSC	LSM	Se	F	LSC	LSM	Se	F	LSC	LSM	Se	F				
Latent period				-				-				-				
line B	α = 1119.75				α = 11.03				α = 91.88							
13 h	39.69	1159.45	324.56		0.44	11.47	3.16		1.83	93.71	5.52					
15 h	−39.69	1080.06	226.59		−0.44	10.58	2.21		−1.83	90.04	3.86					
Latent period				-				-				*				
line 6	α = 1647.23				α = 16.11				α = 79.58							
15 h	−215.73	1431.51	199.04		−2.25	13.86	2.01		9.79	89.37	3.82					
18 h	215.73	1080.06	307.18		2.25	18.36	3.27		−9.79	69.79	6.21					
Line				*				*				-				
Latent period 15h	α = 1388.30				α = 13.19				α = 85.43							
Line B	−150.73	1237.57	160.59		−1.48	11.71	1.54		0.39	85.84	3.60					
Line 6	150.73	1539.03	161.43		1.48	14.68	1.55		−0.39	85.04	3.62					
	Percentage of living embryos								Number of eggs (in thousands)				Number of living embryos after 70 h incubation (in thousands)			
	after 48 h incubation				after 70 h incubation											
	LSC	LSM	Se	F	LSC	LSM	Se	F	LSC	LSM	Se	F	LSC	LSM	Se	F
Latent period				-				-				*				*
line B	α = 90.32				α = 82.95				α = 756.9				α = 640.5			
13 h	2.46	92.79	5.75		1.87	84.82	9.02		27.0	783.9	219.2		28.5	608.9	241.4	
15 h	−2.46	87.86	4.02		−1.87	81.08	6.30		−27.0	729.9	153.0		−28.5	611.9	168.6	
Latent period				*				**				-				-
line 6	α = 61.34				α = 56.76				α = 1113.8				α = 653.6			
15 h	17.14	78.48	5.24		17.06	73.82	5.39		−155.2	958.7	142.4		44.5	699.1	77.6	
18 h	−17.14	44.20	8.53		−17.06	39.69	8.79		155.2	1269.0	231.5		−44.5	608.1	126.0	
Line				*				*				*				*
Latent period 15 h	α = 77.41				α = 71.49				α = 928.9				α = 656.3			
Line B	3.98	81.39	6.14		3.29	74.79	6.78		−104.2	824.7	110.4		−37.0	619.2	125.0	
Line 6	−3.98	73.43	6.17		−3.29	68.20	6.82		104.2	1033.1	111.0		37.0	693.3	125.7	

Se—standard error of the LSM; * *p* ≤ 0.05; ** *p* ≤ 0.01.

**Table 3 animals-13-01428-t003:** Regression predictions for the number of living embryos after 70 h of incubation for common carp (*Cyprinus carpio* L.) breeding lines B and 6.

Breeding Line	Regression Equations	R^2^
B	Y_70_ = −391374.0 + 29762.0x_1_ + 273.1x_2_ + 38955.1x_3_	0.98
6	Y_70_ = 883331.7 − 65267.4x_1_ + 470.6x_2_ − 20527.0x_3_	0.69

Dependent variable: number of living embryos after 70 h of incubation (Y_70_); independent variables: weight of females (x_1_), weight of eggs in g (x_2_), weight of eggs as a percentage of female’s body weight (x_3_); R^2^—coefficient of determination.

## Data Availability

Not applicable.

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
