# Peer review of "The Effect of [(D-Ala6, Pro9NEt)mGnRH-a + Metoclopramide] (Ovopel) on Propagation Effectiveness of Two Breeding Lines of Common Carp (Cyprinus carpio L.) and on Luteinizing Hormone and 17α,20β-Dihydroxyprogesterone Levels in Females during Ovulation Induction"

_animals, 2023, doi:10.3390/ani13081428_

Round 1
Reviewer 1 Report
Nice work, though statistical analysis is a bit too complicated for those who are interested in applicable aspect of this work. Maybe adding PCA analysis would help readers understand the results better. Language needs some polishing.
Author Response
Response to REVIEWER 1
Response to the comments
on the paper ”The effect of [(D-Ala6,Pro9NET)mGnRHa+metoclopramide] (Ovopel) on propagation effectiveness of two breeding lines of common carp (Cyprinus carpio L.) and luteinizing hormone and 17a20ß-dihydroprogesterone levels in females during ovulation induction” by E. Brzuska, M. Socha, J. Chyb, M.SokoÅ‚owska-MikoÅ‚ajczyk, M. Inglot
Comments and Suggestions for Authors
Nice work, though statistical analysis is a bit too complicated for those who are interested in applicable aspect of this work. Maybe adding PCA analysis would help readers understand the results better. Language needs some polishing.
Author’s response:
Thank you very much for reviewing our work. We also thank you for suggesting the use of Principal Component Analysis in our research. In further research on this topic, which we are planning to conduct, we will use this statistical method, which will probably contribute to a better understanding of the results obtained by readers. We have thoroughly read the manuscript and corrected some language issues.
Kind regards,
Reviewer 2 Report
The study tests the effectivity of Ovopel® to induce spawning in 2 carp strains. Investigated parameters are ovulation rate, egg mass, and egg development. Further the study investigates the effect of Ovopel treatment on LH and 7α,20β-DHP levels in blood.
The efficiency of Ovopel to induce spawning in carp is very well documented as well as its effects on hormone levels. There exist also several papers describing differences in propagation effectiveness between carp strains. The study extends this knowledge as it investigates effects of Ovopel on two specific breeding lines which have not been compared until now.
Therefore, the manuscript does not present really new knowledge but complements already existing data. It is therefore considered of low originality and data are mainly useful on a local level for the specific breeding farms.
In my understanding the 2 strains were selected for many generations on specific commercially important traits. It might be possible that this selection procedure did not only change the wanted characteristics but also traits and expression of genes related to reproduction. Thus might lead to the observed differences in egg mass and egg development rates. Investigating or discussing this topic in more detail might increase the importance of the manuscript.
Overall, the manuscript is well organized and clearly written.
Some minor comments
Line 17 streams should be strains
Line 142: 300 g eggs are water hardened eggs?
Line 145/146: The sentence “In addition, the total number of eggs (taking into consideration the weight of 1 carp egg depending on the female BW class … “ is unclear for me.
In my opinion single eggs from each egg batch should be weighed using an analytical balance. Using the mass of a single egg and the total egg mass the total egg number could be calculated
For measurement of hormone levels sample number is small for non-ovulated females of line B (n = 3)
Author Response
REVIEWER 2
Response to the comments
on the paper ”The effect of [(D-Ala6,Pro9NET)mGnRHa+metoclopramide] (Ovopel) on propagation effectiveness of two breeding lines of common carp (Cyprinus carpio L.) and luteinizing hormone and 17a20ß-dihydroprogesterone levels in females during ovulation induction” by E. Brzuska, M. Socha, J. Chyb, M. SokoÅ‚owska-MikoÅ‚ajczyk, M. Inglot
Comments and Suggestions for Authors
The study tests the effectivity of Ovopel® to induce spawning in 2 carp strains. Investigated parameters are ovulation rate, egg mass, and egg development. Further the study investigates the effect of Ovopel treatment on LH and 7α,20β-DHP levels in blood.
The efficiency of Ovopel to induce spawning in carp is very well documented as well as its effects on hormone levels. There exist also several papers describing differences in propagation effectiveness between carp strains. The study extends this knowledge as it investigates effects of Ovopel on two specific breeding lines which have not been compared until now.
Therefore, the manuscript does not present really new knowledge but complements already existing data. It is therefore considered of low originality and data are mainly useful on a local level for the specific breeding farms.
In my understanding the 2 strains were selected for many generations on specific commercially important traits. It might be possible that this selection procedure did not only change the wanted characteristics but also traits and expression of genes related to reproduction. Thus might lead to the observed differences in egg mass and egg development rates. Investigating or discussing this topic in more detail might increase the importance of the manuscript.
Overall, the manuscript is well organized and clearly written.
Some minor comments
Line 17 streams should be strains
Line 142: 300 g eggs are water hardened eggs?
Line 145/146: The sentence “In addition, the total number of eggs (taking into consideration the weight of 1 carp egg depending on the female BW class … “ is unclear for me.
In my opinion single eggs from each egg batch should be weighed using an analytical balance. Using the mass of a single egg and the total egg mass the total egg number could be calculated
For measurement of hormone levels sample number is small for non-ovulated females of line B (n = 3)
Author’s response:
We would like to thank the Reviewer for his effort to thoroughly familiarize himself with our work and for his kind review. Thank you for your comment on the presented research and for pointing out the future direction of research in this area. Including the expression of genes related to reproduction in future research will certainly increase the importance of this topic.
When it comes to some minor comments:
Line 17- the word “streams” has been replaced by “strains”
Line 142 – No, the 300 g of eggs were weighed just after collection from the female, before adding water and fertilization. So, they were not water hardened eggs.
Line 145/146 - in the study by Cejko and Brzuska (2015), the average weight of one carp egg was given, calculated on the basis of a large number of measurements, for each of the five body weight classes of females.
Body weight class of females Average weight of one egg
I >3kg≤5 kg 1.27 mg
II >5kg ≤7 kg 1.29 mg
III >7kg≤9 kg 1.38 mg
IV >9kg≤11kg 1.48 mg
V >11 kg 1.56 mg
In the source data, each female was assigned the appropriate average weight of 1 egg and on this basis the total number of eggs obtained from a particular fish was calculated.
In the opinion of the Reviewer, single eggs from each egg batch should be weighted using analytical balance. Using the mass of a single egg and the total egg mass, the total egg number could be calculated. This method is more precise, but also more laborious.
In future experiments, we will perform calculations using the method suggested by the Reviewer.
The reviewer's opinion that the sample number for non-ovulated females of Line B (n=3) is small is right, but that was the only number we had.
We hope that our work, after making the indicated additions, will be accepted for printing in Animals.
Kind regards,
Reviewer 3 Report
I read with pleasure and interest this article “The effect of [(D-Ala6, Pro9NEt)mGnRH-a + metoclopramide] (Ovopel) on propagation effectiveness of two breeding lines of common carp (Cyprinus carpio L.) and on luteinizing hormone and 17α,20β-dihydroxyprogesterone levels in females during ovulation induc”. I found the work well written and very detailed.
I must admit that in the text is missing informations. In the materials and methods, it should be included how and in what the Ovopel were dissolved before be injected. Moreover, the statistics methods should be better written, including if normality and homoscedasticity were tested and how they were tested. They used parametric and non-parametric tests but the results are presented in mean±SEM, the variables that were analyzed by non-parametric texts they should be presented as median±interquartiles. About the discussion, I found the discussion well written, but it is too short and missing coments about some results, specially about the LH were similar between the ovulated and non-ovulated groups in the both lines. A better discussion is also needed on why they found differences in reproductive performance between the two strains. But in general, the discussion must be rewritten.
Consequently, I believe the work can be published in the journal after some adjust in the manuscript.
Author Response
Response to REVIEWER 3
Response to the comments
on the paper ”The effect of [(D-Ala6,Pro9NET)mGnRHa+metoclopramide] (Ovopel) on propagation effectiveness of two breeding lines of common carp (Cyprinus carpio L.) and luteinizing hormone and 17a20ß-dihydroprogesterone levels in females during ovulation induction” by E. Brzuska, M. Socha, J. Chyb, M. SokoÅ‚owska-MikoÅ‚ajczyk, M. Inglot
Comments and Suggestions for Authors
I read with pleasure and interest this article “The effect of [(D-Ala6, Pro9NEt)mGnRH-a + metoclopramide] (Ovopel) on propagation effectiveness of two breeding lines of common carp (Cyprinus carpio L.) and on luteinizing hormone and 17α,20β-dihydroxyprogesterone levels in females during ovulation induc”. I found the work well written and very detailed.
I must admit that in the text is missing informations. In the materials and methods, it should be included how and in what the Ovopel were dissolved before be injected. Moreover, the statistics methods should be better written, including if normality and homoscedasticity were tested and how they were tested. They used parametric and non-parametric tests but the results are presented in mean±SEM, the variables that were analyzed by non-parametric texts they should be presented as median±interquartiles. About the discussion, I found the discussion well written, but it is too short and missing coments about some results, specially about the LH were similar between the ovulated and non-ovulated groups in the both lines. A better discussion is also needed on why they found differences in reproductive performance between the two strains. But in general, the discussion must be rewritten.
Consequently, I believe the work can be published in the journal after some adjust in the manuscript.
Author’s response:
We kindly thank the Reviewer for familiarizing himself with our work in detail and for his kind, constructive review.
According to the Reviewer's suggestion, in Material and Methods, information on how and in what the Ovopel were dissolved before being injected was introduced (line 131: Ovopel pellets just like dried pituitary glands, were ground in a mortar and then used to prepare a suspension in saline) and information on the statistical methods used was supplemented (line 167-169: Basic statistical tests of the data were carried out using standard procedures included in the Statistica 10 software (StatSoft Polska Sp. z o.o.). For data expressed as percentages, transformations were made using the arcsine function before the analysis.). The results presented as mean±SEM after analysing by non-parametric tests were used in other papers (SokoÅ‚owska-MikoÅ‚ajczyk et al., 2018, https://doi.org/10.4194/1303-2712-v18_2_02; Golshan et al., 2022, https://doi.org/10.1016/j.cbpc.2022.109342; Socha et al., 2023, https://doi.org/10.3390/ani13010105)
Also, as suggested by the Reviewer, the discussion was extended by adding missing contents about some results and future direction of research in this area (lines: 427-428; 431-446; 469-473, 493-499, 547-551 and 560-563).
We hope that after making the indicated additions, our work will be accepted for publication in Animals.
Kind regards,
Round 2
Reviewer 2 Report
I have no further comments